# Design of Low-Cost Modular Bio-Inspired Electric–Pneumatic Actuator (EPA)-Driven Legged Robots

**DOI:** 10.3390/biomimetics9030164

**Published:** 2024-03-07

**Authors:** Alessandro Brugnera Silva, Marc Murcia, Omid Mohseni, Ryu Takahashi, Arturo Forner-Cordero, Andre Seyfarth, Koh Hosoda, Maziar Ahmad Sharbafi

**Affiliations:** 1Lauflabor Locomotion Laboratory, Centre for Cognitive Science, Technical University of Darmstadt, 64289 Darmstadt, Germanyomid.mohseni@tu-darmstadt.de (O.M.);; 2Biomechatronics Laboratory, Department of Mechatronics and Mechanical Systems of the Polytechnic School of the University of São Paulo (USP), São Paulo 05508-030, SP, Brazil; aforner@usp.br; 3Adaptive Robotics Laboratory, Graduate School of Engineering Science, Osaka University, Toyonaka 560-0043, Japan; takahashi.ryu@arl.sys.es.osaka-u.ac.jp (R.T.);; 4Graduate School of Engineering, Kyoto University, Kyoto 606-8501, Japan

**Keywords:** legged locomotion, compliant actuation, pneumatic artificial muscles, mechanical intelligence, control embodiment

## Abstract

Exploring the fundamental mechanisms of locomotion extends beyond mere simulation and modeling. It necessitates the utilization of physical test benches to validate hypotheses regarding real-world applications of locomotion. This study introduces cost-effective modular robotic platforms designed specifically for investigating the intricacies of locomotion and control strategies. Expanding upon our prior research in electric–pneumatic actuation (EPA), we present the mechanical and electrical designs of the latest developments in the EPA robot series. These include EPA Jumper, a human-sized segmented monoped robot, and its extension EPA Walker, a human-sized bipedal robot. Both replicate the human weight and inertia distributions, featuring co-actuation through electrical motors and pneumatic artificial muscles. These low-cost modular platforms, with considerations for degrees of freedom and redundant actuation, (1) provide opportunities to study different locomotor subfunctions—stance, swing, and balance; (2) help investigate the role of actuation schemes in tasks such as hopping and walking; and (3) allow testing hypotheses regarding biological locomotors in real-world physical test benches.

## 1. Introduction

Legged locomotion poses a challenging, hybrid, nonlinear, and highly dynamic problem [1]. Although humans and terrestrial biological locomotors demonstrate efficient and robust legged locomotion, contemporary bipedal robots struggle to achieve rapid, robust, and untethered dynamic gaits in natural environments over extended distances [2,3,4,5]. This challenge stems from an incomplete understanding of various locomotion aspects, including mechanical design, actuation, and control. Understanding biological locomotor systems can enhance legged robots and assistive systems. In this regard, human experiments, data analysis, and model development have been common approaches taken so far. However, the challenge persists in bridging the gap between models and real-world scenarios, questioning the sufficiency of simulations for validating identified biomechanical concepts (e.g., mechanisms, motor control) of biological locomotor systems. The main challenges with simulation models for predicting human movement and understanding motor control and biomechanics revolve around accurately replicating the complexity of human motion, motor control, and the biomechanical properties of tissues and joints. These challenges include the dynamic and nonlinear nature of human movements, the integration of sensory feedback for motor control, and the variability in individual biomechanical properties. Developing bio-inspired robots offers a promising approach to addressing these issues by creating systems that mimic the structural, functional, and control aspects of human motor systems. Studies in this field aim to develop robots and software platforms that can replicate human movements within human-centric environments, leveraging advances in modeling and simulation to assess and control aspects such as fatigue outside of clinical settings [6,7] or to mimic mechanical behavior and motor control [8]. Therefore, it has become crucial to develop cost-effective, accessible, and modular test platforms with sufficient degrees of freedom (DoFs) both in mechanics and actuation for robotic and biomechanics research in locomotion.

To design test platforms for achieving adaptable and efficient locomotion, suitable actuators are indispensable, particularly in providing force demands, e.g., in loaded, highly dynamic, or explosive movements. Besides the need for substantial force, there is a strong inclination toward variable impedance [9,10]. For example, human-running biomechanics exhibit significant variations in effective leg stiffness corresponding to changes in speed [11] and ground stiffness [12]. Electric motors (EMs) have a power mass density exceeding that of biological muscles (up to 7 kW/kg, 3–5 kW/kg compared to a maximum of 0.3 kW/kg for muscles) [13], with their high power primarily available at high speeds and accompanied by relatively low torque compared to muscles. While an increased gear reduction can augment torque density, it simultaneously elevates passive impedance in actuators (inclusive of reflected inertia, friction, and damping), thereby limiting bandwidth and compromising transmission transparency. As an alternative, researchers have explored impedance modulation through torque feedback and full-state feedback [14] or the employment of series elastic actuators [9,15,16]. Other approaches have included the design of dual actuators that enhance stiffness modulation [17,18] and tunable series elasticity, which have shown promising results when applied in manipulation and mobile robots [10,19,20,21]. Efforts to enable compliant actuation using EMs without sacrificing torque density are ongoing, but a winning design replicating the qualities of biological actuators has not yet been achieved.

In exploring various actuator design methods, the integration of EMs and pneumatic artificial muscles (PAMs) in electric–pneumatic actuators (EPAs) [22] offers a unique avenue for investigating the interplay of motor control and morphology in robotic systems. PAMs have been applied as actuators in a diversity of robots and exoskeletons due to their intrinsically low passive impedance, relatively high force density [23,24,25,26,27], and adjustable compliance for enhancing motion efficiency and robustness [28]. Previous studies on EPA-based hopping robots have demonstrated the benefits of combining passive and active impedance, showcasing improvements in efficiency, robustness, and controller generalization [28,29,30,31]. This approach leverages the body as a computational resource, incorporating intelligence into the mechanical design, particularly through the use of soft materials, termed “mechanical intelligence” [32,33,34,35,36]. Moreover, incorporating PAMs into the actuation system facilitates their seamless integration into various configurations within the robot structure, such as mono-articular or bi-articular arrangements [31]. Given that PAMs share comparable behavioral dynamics with biological muscles [37,38], this integration provides a unique opportunity to explore and understand the functions of human muscle–tendon units in the context of robotic applications. Consequently, the use of PAMs through EPA provides a framework to develop reverse engineering tools, offering a means to explore and validate novel findings about biological muscles [39].

In acknowledging the current gaps in understanding the mechanical design, actuation, and control of legged biological locomotors, this paper focuses on the design and development of a series of robots actuated by a combination of EMs and PAMs. In our previous works, we developed three hopper robots actuated by EPA; see Figure 1. The first robot, MARCO-Hopper II, was a two-segmented robot actuated by a single motor augmented by a parallel PAM, designed for one-dimensional hopping [29]. Later, EPA-Hopper I was introduced, featuring a human-sized design with increased DoFs [21,30]. Finally, EPA-Hopper II extended the previous design with a foot and ankle joint [31]. Throughout these iterations, a common characteristic persisted: the limitation of hip joint movement in space and the absence of a trunk, restricting the examination to stance and partly swing locomotor subfunctions [1]. Expanding on the research of three hopping robots, this study introduces two new robots: a monoped jumper with additional DoFs capable of two-dimensional motion and a biped walker, which represents our initial exploration into bipedal locomotion (Figure 1). The total cost of constructing a single leg with the trunk is below EUR 3000 less than that of alternative robotic designs [5]. These robotic platforms are accessible tools with multi-actuation EPA technologies that can facilitate the exploration of different aspects of locomotion and allow for a deeper exploration of the complexities inherent in legged locomotion.

## 2. Overview of Previous EPA-Robot Developments

This section provides a detailed overview of EPA actuation and previous EPA robots. It begins with an outline of the EPA technology used in these robots, including the strengths and weaknesses of each actuator and how they complement each other. Following this, it briefly discusses the evolution of EPA robots and some of their contributions. Finally, the previously implemented control strategies in EPA robots are concisely reviewed.

### 2.1. Electric–Pneumatic Actuation

Electric motors (EMs) are renowned for their high-density power, offering precise position and torque control across a broad spectrum. They excel in continuous operation, exhibiting a remarkable torque-velocity range [13]. However, legged locomotion tasks that involve handling impacts necessitate efficient actuation across a diverse torque and velocity range and entail close interaction with uncertain environments. Within this context, EMs encounter limitations, and their inherent strengths are somewhat compromised [22,40]. Conversely, PAMs present an alternative with adaptable compliance and a force–length relationship comparable to biological muscles [41]. Their high force-to-weight ratio, coupled with low weight and inertia, positions PAMs as an attractive actuation technology in bio-inspired and human-safe robotics [42,43]. Nevertheless, the non-linear nature of the pressure–force relationship and the low bandwidth of PAMs make it difficult to control PAM-actuated robotic systems [22,44]. Consequently, stability in solely PAM-driven systems for legged robots is typically achieved through manual tuning of control parameters rather than through analytical methods [1]. This approach often leads to suboptimal performance and can make it difficult to generalize the control strategies to different scenarios or environments.

Drawing inspiration from the functional performance and neuromechanical control observed in biological muscles [45], the adoption of a compliant, redundant actuation system with ample control bandwidth has emerged as an ideal approach for facilitating legged locomotion. Based on this insight, the integration of EMs and PAMs addresses the shortcomings inherent in each actuation system through a combination of their respective advantages for locomotion tasks [22,46]. PAMs introduce adjustable mechanical compliance, enhanced adaptability, and redundancy in design morphology—features less attainable with EMs (see a detailed comparison in Table 1). However, PAMs are also characterized by their response delays and a lack of precise control capabilities. Adjustable mechanical compliant actuation design and precise control are the main advantages that complement each other to produce a new actuator with higher performance, efficiency, and adaptability for designing a legged robot. Such a holistic approach in employing an electro–pneumatic actuation (EPA) system guarantees a compliant actuation framework, characterized by high power density alongside enhanced control over position and torque, marking a significant stride in robotic locomotion technology.

### 2.2. Previous Iterations of EPA Robots

Before delving into the specifics of the recently developed EPA Jumper and EPA Walker iterations, here we provide an overview of previous developments in EPA-based robots, as also shown in Figure 1.

*MARCO-Hopper II:* The MARCO-Hopper II marked the initial entry in the EPA-robot series, featuring a robotic structure comprising two metal segments resembling the human thigh and shank in the sagittal plane [47]. These segments were connected by a revolute joint resembling the knee joint. Linear bearings guided the hip and foot joints, constraining motion to vertical translation, resulting in a one-dimensional hopping motion. This robot utilized an off-board EM, which in [29], was augmented by a single PAM on the knee joint to investigate the effect of adaptable parallel compliance on hopping efficiency.

*EPA-Hopper I:* The first robot designed and developed for exploring EPA technology for bio-inspired legged locomotion was EPA-Hopper I. In contrast to its predecessor, this iteration represented a robotic leg that resembled the size of a human leg, in which hollow carbon fiber tubes were utilized as segments. It also featured increased DoFs via the addition of an actuator at the hip joint in the setup, which also enabled the foot to move in 2D. This enhancement facilitated the exploration of a more realistic hopping motion [21,30]. Actuation in this robot involved two EMs at the hip to actuate hip and knee joints and two PAMs on the knee joint to provide adjustable compliance to the leg (see Figure 1).

*EPA-Hopper II:* The progression culminated in the creation of EPA-Hopper II, an extension of EPA-Hopper I featuring the addition of a 3D-printed foot and ankle joint [31]. Similar to its predecessor, this robot employed two EMs for actuation but incorporated different mono-articular and bi-articular PAMs in its structure. Drawing inspiration from the human leg musculoskeletal system, where joint compliance increases from the hip to the knee and then to the ankle [48,49,50], the EPA actuation architecture in this robot featured an EM at the hip joint, a combination of EM and PAMs at the knee joint, and solely PAMs at the ankle joint.

### 2.3. Control Approaches

Several controllers have been proposed to facilitate the generation of energy-efficient and stable hopping motions while simultaneously exhibiting robust responses to perturbations. In the study by Kalveram et al. [51], different energy management methods were assessed for achieving stable hopping in the presence of losses and perturbations on an actuated prismatic leg system (MARCO-Hopper). The findings revealed that injecting a predetermined amount of energy greater than the system losses during a hopping cycle resulted in stable, human-like hopping. However, the absence of a feedback signal in this control scheme renders it sensitive to uncertainties and disturbances, requiring precise parameterization for proper functionality. Subsequently, in the segmented MARCO-Hopper II, a virtual model control (VMC) approach [52] was employed, emulating a virtual spring between the hip and foot. This VMC strategy demonstrated effectiveness in producing stable, human-like hopping movements [47]. The control framework was also extended and compared with feed-forward control in terms of hopping stability and perturbation rejection in [47]. Inspired from the positive force feedback [53] observed in bouncing gaits [54,55], a neuromuscular reflex-based controller was implemented on GURO monopod and bipedal robots. Simulation and experimental results showcased the generation of hopping through the utilization of positive force and length feedback reflexes [56]. However, the performance of these methods is dependent upon the level of detail considered in the controller implementation, and determining the appropriate parameters involves an exhaustive search [55]. Inspired from the concept of positive force feedback [54], the force modulated compliant (FMC) controller was introduced [57]. The FMC controller simplifies the complexity associated with previous neuromuscular controllers, offering a simplified version of force feedback with fewer tuning parameters. Demonstrating success in both simulation and experiments, the FMC controller has proven effective in generating various hopping motions with different frequencies and heights across different EPA robots, ranging in complexity from MARCO-Hopper II to EPA-Hopper II [29,30,31].

The implementation of the FMC controller involves dividing hopping into two distinct phases: stance and flight. During the stance phase, FMC control is applied to the knee (referred to as FMCK), while the hip EM is deactivated. The torque (τ) applied by the FMC control is $\tau =KF(\phi -\phi_{0})$, where *K*, *F*, ϕ, and $\phi_{0}$ are normalized stiffness, ground reaction force (GRF), joint angle, and rest angle, respectively. This controller introduces virtual stiffness to the knee joint, modulated in real-time by the GRF, thereby inducing a nonlinear spring-like behavior. Notably, the role of GRF as proprioceptive feedback for locomotion control finds support in prior biomechanical studies [58,59,60]. The functionality of this bio-inspired controller can be interpreted as a simplified reflex control in which the muscle and muscle force are replaced by a spring and the leg force (measured by GRF), respectively. In the flight phase, position control is implemented for both the hip and knee motors to ensure consistent touchdown during each hop.

## 3. EPA-Jumper and EPA-Walker Design

In this section, we present detailed information on the mechanical design, actuation, and electronics of the EPA leg and the trunk. The modular design of the EPA leg allows for versatile applications. This robotic leg can operate independently as a monoped, known as the EPA Jumper. Alternatively, through symmetric duplication of the EPA legs, connection of the two trunks with aluminum plates, and integration of the additional EPA leg in series in the EtherCAT bus, the design can be effortlessly transformed into a bipedal configuration, as illustrated by the EPA Walker in Figure 1. This modular approach facilitates seamless extension and allows for future design enhancements for both the EPA Jumper and Walker.

### 3.1. Mechanical Design

This section provides a detailed overview of the mechanical design of the newly developed 3-DoF EPA leg with a trunk. The focus is on the design aspects of the trunk, hip, knee, and ankle, along with their interconnections to each PAM. Emphasis is placed on overcoming challenges encountered in previous versions of robots while maintaining a human-like weight distribution in the design and using low-cost and readily available materials and methods such as 3D printing. For EPA-robot development, we used *Prusa i3 MK3S* for printing different parts (e.g., joints and the foot) with *PLA*. Additionally, we describe the actuation scheme implemented in this robot and draw comparisons with its predecessors. The robot’s complete schematic is presented in Figure 2, and its components are listed in Table 2.

#### 3.1.1. Trunk

To achieve a combination of enhanced rigidity and optimized weight distribution, the trunk’s main structure is made of 3 mm thick aluminum plates. The trunk’s two-level architecture effectively integrates all the robotic system’s electronics, sensors, and actuators (Figure 2). The first level positions ten on/off valves (SMC SYJ3320) for effortless accessibility, accompanied by their corresponding pressure sensor (SMC PSE530). Furthermore, attachment points for connecting PAMs with designed lever arms are strategically located at the outermost sides. The second level houses dedicated boards for pneumatic and motor actuation, with further specifications detailed in the subsequent section. The two horizontal plates are securely fastened to each other and connected to the leg through two vertical plates. These vertical plates are directly affixed to the hip joint through the hip stator and ball bearing (6304-RS) to properly center the leg and distribute its forces (Figure 2). Additionally, these vertical plates are specifically crafted to incorporate three steel ball casters, allowing for free-wheeling motion (not shown in Figure 2). By employing these ball casters, the robot’s structure and movement can be confined to the sagittal plane within a frame, akin to the design of the BioBiped robot [16]. The EPA-Jumper robot is the combination of one leg and the trunk. The EPA-Walker robot was developed by connecting two EPA-Jumpers through two horizontal aluminum plates at their trunks, as shown in Figure 3b. The size of the trunk width for EPA-Walker is adapted to match human body characteristics. The designed mechanism is rigid to support 2D movement, which can be extended by adding elasticity in the lateral plane for moving in 3D in the future.

These design choices align with the goals of replicating a more balanced distribution of segment mass and elevating the center of mass (CoM). These are key features in achieving comparable system dynamics for human-sized robotics. Future mechanical design iterations will specifically target the reduction of the emergent disparities in these key features.

#### 3.1.2. Hip

In alignment with our prior design, we employ a directly driven EM for hip joint actuation, while the knee joint is actuated utilizing a rope-and-pulley system. This configuration serves a threefold purpose. Firstly, placing both EMs at the hip joint reduces the leg inertia. This reduction contributes to both energy efficiency by facilitating the movement of lighter distal segments and robustness, as there is less direct impact on the EMs and associated electronics. Secondly, opting for a rope-and-pulley system rather than a gearbox mitigates friction in the transmission chain [61]. This setup facilitates torque control implementation through motor-current sensing [30]. Thirdly, the utilization of a rope-and-pulley system for transmission facilitates the incorporation of a PAM in series with the knee motor, thereby establishing a series of elastic actuation for the knee joint [21]. This serial arrangement of compliance serves to decouple the motor’s inertia from the load, imparting advantages in terms of shock absorbance and energy buffering, as highlighted by [62,63].

In the new iterations of EPA robots, the coaxial EMs are oriented facing each other by the use of a ball bearing (6001-RS). Centering the rope-and-pulley system effectively mitigates undesired lateral forces, a crucial consideration to preventing adverse effects on 3D-printed parts. Another noteworthy modification entails the increased thickness of segments, specifically carbon-fiber tubes. This adjustment has been implemented to proactively address potential challenges arising from increases in both the size and weight of new robots.

To address the challenges of high torque and impact, the knee pulley at the hip level has been manufactured using a carbon-induced 3D printer, reinforcing the part for robust performance. All other components at the hip are 3D printed. Additionally, ball bearings (6304-RS) are employed to couple the knee motor to the trunk, providing seamless mechanical connectivity.

#### 3.1.3. Knee

The knee joint configuration comprises three main 3D-printed components and incorporates a ball bearing. It freely allows movement in flexion and extension directions but incorporates mechanical constraints to prevent overextension, yielding a range of motion similar to that of the human knee joint. Notably, in the latest iterations of EPA robots, the knee design has undergone modifications compared to its predecessor. A key enhancement involves centralizing the torque transmission from the EM motor located on the hip to the knee through a rope-and-pulley system, effectively mitigating undesired forces. The knee-to-hip pulley transmission ratio has been set to 2.5, hence allowing for the higher knee accelerations needed for high-frequency jumping and explosive tasks. For joint coupling, a ball bearing (6304-RS) is employed, and all components of the knee joint are produced with 3D printing.

#### 3.1.4. Ankle

In line with previous iterations of the EPA robot, the ankle remains a single DoF joint, actuated solely by PAMs (TIB, SOL, and GAS), allowing for movement primarily in the sagittal plane—enabling dorsiflexion and plantarflexion. Affixed to the ankle is a curved, human-shaped foot designed to mimic the contours of the human foot. To enhance shock absorption during initial ground contact, a rubber sheet is incorporated beneath the foot.

The curved configuration of the foot, as opposed to a pointed foot, promotes a more natural gait and has demonstrated greater energetic efficiency [64,65]. The connection between the foot and the leg is facilitated by a ball bearing (6202-2Z), and both the foot and ankle components are fabricated using 3D-printing technology.

### 3.2. Actuation

In line with its predecessors, the latest iteration of EPA robots in this study is actuated with two EMs situated at the hip joint, one driving the robot hip and one using a rope-and-pulley system for actuating the knee joint. Given the new iteration’s increased weight, the previously used E8318-120KV Hymotor in the EPA-Hopper robot is substituted with GIM8115-9 EM. These EMs offer significantly higher nominal torque (13Nm) and a greater torque constant (3.5Nm/A). The increased torque constant lowers battery consumption, enabling longer experiment durations. Additionally, the motor’s compactness, owing to its built-in driver, and its geometric design facilitate seamless mechanical integration.

In addition to EMs, each joint is complemented by PAMs designed to mimic the corresponding muscles in the human body (Figure 3a). The number of PAMs integrated into the new iteration has expanded from four in EPA-Hopper II [31] to nine (Figure 3a). Six mono-articular PAMs encompass both extension and flexion at each joint, while three bi-articular PAMs—gastrocnemius (GAS), hamstring (HAM), and rectus femoris (RF)—play a crucial role in inter-joint energy transmission, significantly contributing to the robot’s balancing, stabilization, and overall performance. Every PAM features both a fixed and an adaptable attachment, facilitated by a tension belt buckle, allowing for the adjustment of its rest length. Additionally, various extensions for the lever arm ratios of the PAMs at the trunk and knee level are considered through multiple exchangeable holds, enhancing the versatility and adaptability of the robotic system.

The PAMs utilized in the robot structure are custom made, comprising an elastic rubber inner tube enclosed with an expandable polyester shell. Precise control over the inner tube pressure is achieved using on/off valves (SMC SYJ3320) as opposed to continuous valves (PVQ-31, SMC, Misumi, Tokyo, Japan) used in previous iterations. This selection of valves for the newly developed robots is motivated by two primary factors. Firstly, on/off valves possess the capability to both pressurize and depressurize. Secondly, the chosen valves contribute to a reduction in overall weight, as they are more lightweight than are continuous valves.

### 3.3. Electrical Design

The electronics architecture consists of multiple layers. At its core, a high-level control layer operates within a Simulink Real-Time framework on a target PC. This controller communicates with the slaves via an EtherCAT bus. The low-level control layer comprises three slaves, responsible for actuating and sensing the motor, the PAMs, and the GRF plate. A simplified diagram is shown in Figure 4.

#### 3.3.1. High-Level Control Layer

For real-time control and monitoring of the robot, we utilize the MATLAB Simulink Real-Time Toolbox (xPC) as the software framework. To ensure fast and robust communication, EtherCAT (Ethernet for Control Automation Technology) serves as the chosen field bus system for our robots. This setup involves two computers: the xPC target computer and a host computer. The host computer facilitates direct control and monitoring of the robot, including all slaves, through the Simulink environment. The xPC target computer processes data, determining commands for various slaves. This configuration allows for real-time parameter tuning and debugging the robot control via the Simulink interface while maintaining a high running frequency of 1kHz.

#### 3.3.2. Low-Level Control Layer

*(A) Motor Board*: The control of hip and knee motors, along with the readings of the motors’ responses in terms of currents, position, and velocity, is managed by an Arduino board. The communication between the motors and the micro-controller is facilitated by an MCP2515 CAN-BUS controller and transceiver module. This module communicates with the micro-controller through an SPI bus while establishing a connection with the hip and knee motors through a CAN bus. In the context of CAN bus usage, it is crucial to address the potential for noise to enter the system. To ensure robust performance, careful consideration must be given to appropriate cable lengths and the incorporation of termination resistors. These measures are essential for mitigating potential issues associated with noise and maintaining the reliability of communication within the system.

*(B) PAM Board*: For the precise control of SMC SYJ3320 valves and accurate readings from SMC PSE530 pressure sensors, we have chosen the Arduino Mega due to its compatibility with 5 V analog inputs, aligning with the signal voltage of the pressure sensors. Valve control is managed through a custom-built PCB featuring Silicon N-Channel MOSFETs K4017 to activate the solenoids in the valves. This PCB incorporates additional components, including base resistors of 150 ohm and 10 k ohm, along with a diode implemented to protect against voltage spikes, enhancing the overall reliability and durability of the system.

*(C) GRF Board*: For precise measurement and detection of GRF, we utilize piezoelectric Kistler force plates (type 9260AA). These force plates provide three-dimensional readings of force, including the center of pressure. The integration of GRF plate readings into the system is achieved using an Arduino board, which directly interprets analog signals from the force plate. The Arduino board is converted into an EtherCAT slave using Shield EasyCAT. The embedded code in the Arduino ensures efficient reading of the four force sensors on the GRF plate, transmitting the data to the host PC. These sensor data are then used for gait detection, control, and data analysis. In later stages of development, the GRF plate will be exclusively used for testing with high-precision measurement to assess the functionality and controllability of the robot. Instead, an insole sensor will be adopted to create a standalone robot capable of ground locomotion. In addition to GRF measurements, this board also oversees the monitoring of an emergency switch. Activating the emergency switch disables all communications, thereby turning off actuation. This safety feature ensures prompt response in critical situations, enhancing the overall reliability and safety of the system.

It should be noted that the GRF plate operates within a power voltage range of +10 to +30 V, and thus both the Arduino and force plate are powered by an external +12 V power supply or battery. This power supply serves as the energy source for all system boards, valves, and pressure sensors.

## 4. Results

In this section, following a detailed comparison of the mechanical characteristics between the designed EPA robot and humans, we present the results of the preliminary hopping tests with the EPA Jumper. Finally, we present and examine an overview of the hopping patterns achieved across various iterations of EPA robots.

### 4.1. Mechanical Properties: Robot vs. Human

We evaluated the mechanical characteristics of the EPA Walker, including segment lengths, weight distribution, and inertia. Measurements of segments’ length and weight were obtained from the physical system, while inertia values were approximated using the CAD design in Autodesk Fusion 360; see Table 3. The EPA Walker was compared to a human model with a height of 1.65m and a weight of 60kg. The human model characteristics were estimated using information from [67].

In the comparison of the segments’ length between the robot and the human, a consistent proportion was noted for the shank and thigh. However, there were differences in the foot and trunk segments. In regard to the segments’ weight, a fairly similar weight distribution was replicated, with only a slight deviation in the shank’s weight. However, the same similarities were not replicated in the system inertia. On the shank and thigh levels, a consistent inertia ratio was achieved. Meanwhile, the foot and trunk need modification to approach a human-like inertia ratio.

### 4.2. Preliminary Results of EPA Jumper

To assess the mechanical structure and performance of the recently developed EPA Jumper, we conducted preliminary hopping tests with the robot. Despite the increased complexity of the new robot, we employed the FMCK control scheme as utilized in previous EPA robots to achieve hopping [29,30,31]. The values for control parameters and PAM pressures were manually tuned and are presented in Table 4.

Throughout the experiments, the robot’s motion was confined to the sagittal plane, limiting lateral movement. As the only controller in the stance phase is implemented on the knee joint, we set the knee PAMs’ pressure to zero to minimize extra effects on knee actuation and focus on the extendability of the stabilization approach with minimal control. Hip muscles were pressurized to approach the upright trunk condition of experiments with previous EPA robots. The leg configuration in the flight phase was borrowed from previous studies and slightly adapted. The experimental outcomes are illustrated in Figure 5. Despite the introduction of a trunk and the necessity for balance maintenance, the results demonstrate that employing the same FMC control approach and ankle PAM configuration, along with utilizing additional PAMs at the hip joint, proved effective in attaining repetitive hopping patterns. The consistent periodic patterns were observed in the knee and hip joints as well as the GRF. The robot moved 26cm backward within eight hops observable from COP curves. It is worth noting that in our additional preliminary tests, we achieved two and four non-repetitive hops without the inclusion of hip PAMs and by solely relying on mono-articular hip PAMs, respectively.

### 4.3. Hopping Performance with Different EPA Robots

To evaluate how much the body complexity and incorporation of a more human-like structure affect hopping performance, we compared hopping patterns obtained from the different EPA robots shown in Figure 1, except the EPA Walker. Appendix A demonstrates the performance of these robots. Normalized GRF and hip displacement are illustrated in Figure 6 as representatives of hopping performance. This figure serves as a demonstration of the adaptability and extendability inherent in the EPA design. Examining the GRF patterns, we can observe a progression toward a more human-like hopping motion as mechanical constraints are released (from the foot and then the hip), additional degrees of freedom are introduced to the ankle and hip joints, and additional body segments, such as the foot and trunk, are incorporated. Notably, the GRF pattern of the EPA Jumper stands out as the smoothest and most human-like despite no changes being made to the controller. Interestingly, normalized hip joint excursion remains relatively consistent across different robots, with MARCO-Hopper-II exhibiting the minimum and EPA-Hopper I achieving the maximum hopping height. Furthermore, EPA Jumper and EPA-Hopper I demonstrate the smoothest (most human-like) hip displacement patterns.

## 5. Discussion

Bio-inspired experimental robotic setups take a step beyond simulations to deepen our understanding of the functionality of biological mechanisms and to verify identified human motor control concepts in the real world. Despite advancements in modeling approaches, e.g., using deep reinforcement learning methods to explore human locomotion and predicting movements in real environments, modeling human motor control accurately is still a substantial challenge [69]. Bio-inspired robotic systems could address these challenges and facilitate understanding human movement, offering insights into developing intelligent robotic systems that can adapt and interact in complicated, dynamic environments [6]. However, their widespread usage is hindered by high costs, complex designs, reliance on inaccessible materials, and restrictive actuation technologies which are highly dissimilar from human muscular actuation. To overcome these challenges and promote easy access to such setups, this work introduces an innovative series of bio-inspired low-cost robots constructed with off-the-shelf components and 3D-printed parts. These robots employ EPA technology, enabling mechanical and morphological adaptability through adjustable intrinsic compliance to replicate muscle-like actuation as well as high controllability and energy density. These modular setups offer a versatile platform for exploring a wide range of bio-inspired locomotion patterns and provide a means to investigating the interplay between body morphology and control strategies.

### 5.1. Morphological Similarity and Modularity

The newly developed EPA robots introduced in this study were designed to possess a mass distribution and inertia ratio closely resembling those of their biological counterparts, the human body. This meticulous replication of mechanical properties enables the robots to exhibit locomotion patterns more similar to natural human movement, positioning them as promising tools for exploring the fundamental principles of biological locomotion. However, certain disparities in the ratio of inertia among some segments still exist, requiring further considerations for the next iterations. Addressing these disparities will be a focus of our future work.

In addition to maintaining mass distribution and inertia similar to human legs, EPA design provides further access to mimicking the morphological properties of biological locomotors. Utilizing EPA technology allows for a hybrid variable impedance actuation that can mimic biological actuation while addressing the limitations of robots solely driven by PAMs. The integration of EMs and PAMs holds the potential to enhance controllability and to address the limitations inherent in each actuator type. More importantly, the combination of EMs and PAMs enables a versatile array of configurations. PAMs can be employed in parallel/series to EMs or in bi-articular arrangements. Parallel PAMs to EMs could reduce power or torque requirements, provided their tuning aligns appropriately. Another feasible arrangement involves a PAM crossing two joints, functioning as an energy exchanger. Bi-articular configurations have proven advantageous in terms of contributing to posture and body balance [70], increasing energy efficiency [71], and simplifying control tasks [16]. Our preliminary experimental results with EPA Jumper demonstrate the effects of the passive dynamic behavior provided by PAMs (in ankle and hip joints) to generate repetitive hops without demanding any extra effort for active control design. Owing to the leg’s muscle-like PAMs, several repetitive hops can be achieved with a switched-off hip motor at the stance phase. The contribution of even passive (with fixed pressure) mono-articular and bi-articular PAMs to stabilize trunk and body posture can be identified by an increase in the number of repetitive hops. Therefore, inspiration from human musculoskeletal architecture, which is easily accessible via EPA design, could open new doors in learning from morphological computation (more details in the next section).

Both the EPA Jumper and EPA Walker are equipped with six mono-articular and three bi-articular PAMs per leg, emulating the main lower-limb group muscles. Moreover, an additional PAM could be placed in series with the knee motor, emulating a muscle–tendon unit. Incorporating PAMs in series with EMs provides the system with the capability to absorb shocks and withstand impacts and increases robustness against perturbation, as shown in [21]. Additionally, this serial configuration can enhance energy efficiency in cyclic tasks.

We can leverage PAMs as passive elements with adjustable compliance, which can complement EMs to reach optimal solutions (e.g., regarding energy efficiency) for different tasks (e.g., hopping, walking, running) or gait conditions (e.g., hopping frequency, walking speed) while their rest length can be manually tuned. In addition, PAMs can function as actuators by real-time pressure adjustments. This high degree of modularity allows researchers to customize the robots’ behavior to study various aspects of human locomotion.

The human neuromuscular system is highly complex, making its operational principles challenging to comprehend solely through movement outcomes and simulation models given the limitations in directly altering the fundamental properties of the human body for verification [72]. However, developing physical blueprints of the biological body could serve as a means to validating biomechanical theories [73]. In this regard, the modular structure of EPA robots has the potential to facilitate the reverse engineering of human biomechanics and motor control by providing a framework for systematic exploration and testing.

### 5.2. EPA Actuation and Control Embodiment

The human neuromuscular system exhibits profound complexity, evolving from simpler neuromuscular modules in infancy to a more intricate structure in adulthood, integrating multiple motor control levels [72]. This developmental trajectory, marked by increasing modularity and connectivity, mirrors the principles of nature’s gradual progression from simple to complex systems [74]. This evolution involves multiple levels of the motor infrastructure, from the intrinsic rhythmicity measured at the level of individual muscle activities to the level of muscle synergies and bilateral inter-muscular network connectivity [72]. In constructing bio-inspired robots, embracing this principle of starting with basic structures and progressively adding complexity can offer insights into underlying control mechanisms. This approach resonates with the templates and anchors concept [75], emphasizing the utility of simple yet robust control solutions as foundations for complex behaviors.

In this work, the newly developed EPA Jumper was subjected to some preliminary tests in which periodic hopping was achieved. Despite the EPA Jumper’s more complex structure compared to its EPA-based predecessors, the generation of stable hopping patterns did not result from a more complex controller. Rather, it was attributed to the modularity inherent in the EPA-based robot design. By adjusting the PAMs’ pressures and rest lengths within the EPA design, a portion of the control problem can be effectively delegated to the body dynamics. This approach allows the body to function as a computational resource, reducing the overall control effort, which aligns with the concept of brainless locomotion [76,77,78] incorporating the morphological computation paradigm [79].

Control embodiment through morphological computation and employing compliant elements introduced in prior studies [15,80,81] has been addressed in our research on EPA-based robots, spanning from 1D hopping in MARCO-Hopper to 2D hopping with EPA-Hopper. Appendix A shows how this concept has evolved from hopping with one degree of freedom (DOF) to four DOFs, with consideration given to trunk balancing in the sagittal plane in the EPA-Jumper robot. The preliminary tests on the EPA-Walker robot presented in this video support the mechanical durability of this bipedal robot to handle expected impacts and forces, which needs further investigation to generate a stable gait. Up to now, our focus has remained on optimizing the EPA design to accomplish the task with minimal changes to the controller across different levels of mechanical complexity. For instance, EPA-Hopper II, despite an additional DoF and foot segment compared to EPA-Hopper I, achieved stable hopping without any alterations to the controller parameters, highlighting the robustness facilitated by the capabilities of PAMs in the EPA design [30]. Building on this, our work with the EPA Jumper employed the same controller and relied on hip PAMs (HAM, RF, IL, and GLU) to stabilize the balance task, achieving hopping without an increase in the complexity of the control strategy. This accomplishment underscores the remarkable adaptability enabled by EPA actuation, allowing for more complex robots such as the EPA Jumper to employ a consistent and straightforward control strategy across diverse, dynamic systems. Furthermore, the same control applied on a such body of increased complexity yields more human-like hopping patterns, as shown in Figure 6. This minimalistic control and the utilization of body intelligence to generate biological behavior support previous studies [15,82,83].

While the EPA Jumper demonstrated successful repetitive hopping, this performance was limited to only a few consecutive hops (see Figure 5). One indicator of the non-stationary hopping pattern is the undesired backward shift of the center of pressure. Unlike human hopping, the robot knee exhibits limited bending followed by a large extension at the stance phase (opposite to the flight phase), which needs to be improved to increase joint synchronicity and hopping stability. These observed behaviors stem from the fact that the parameters chosen for the experiments were not the result of optimization but rather rough hand-tuning. Addressing these unwanted behaviors could involve tuning the controller parameters and adjusting PAM pressures. It is remarkable that the performance described here was achieved without hip motor control in the stance phase. Therefore, one substantial step is to expand the control framework to incorporate the hip joint and utilize PAM adjustments to enhance trunk balancing to achieve more stable, robust hopping patterns comparable to human locomotion. Still, the hip controller could follow the minimalistic design approach to benefit from the system’s dynamic behavior. One potential approach would be implementing the FMC controller, which was initially developed for predicting the hip torque for balancing in locomotion [57,84,85,86]. This way, we will keep the essence of control embodiment and employ morphological computation using the EPA infrastructure.

### 5.3. Reverse Engineering the Biological Locomotor System

The overarching goal of developing EPA technology for the design and control of legged robots is to approach the human musculoskeletal system and enable the implementation of human-like motor control. We aim to offer a cost-effective and versatile platform for probing biomechanical theories in legged locomotion. Altering a mechanism in the human body (e.g., sensory input or muscle) for testing a hypothesis is almost impossible, but it can be easily tested by adapting the representative component in EPA robots. Analyzing the effect of serial and parallel compliance on VAS muscle [21] or investigating the roles of VAS and GAS muscles in hopping [31] are examples of such studies with previous EPA robots. Such approaches also highlight how a robot’s physical structure can inherently manage aspects of locomotion, reducing the need for complex control algorithms. By leveraging mechanical properties such as elasticity, robots can achieve more natural, energy-efficient, robust, and agile movement, mirroring the adaptability seen in biological systems [87]. These principles are being increasingly integrated into robotic designs to enhance their interaction with complex environments, drawing inspiration from the sophistication of biological locomotor systems [88].

### 5.4. Outlook

On the mechanical side, future work will involve the design of new iterations of the EPA robots to better reflect the same weight and inertia ratios of humans while still preserving the cost and modularity of the robot. On the biomechanical and control side, we aim to unlock the complete potential of the EPA approach by expanding its application to different forms of locomotion under both normal and perturbed conditions [70]. Building upon our previous work, we seek to gain a deeper understanding of the adaptations necessary in both EPA design and control mechanisms to match different locomotor subfunctions [1] seamlessly. The newly developed EPA robots extend the possibility of investigating locomotor subfunctions from stance to swing and balance subfunctions. Additionally, we aim to extend the EPA approach to encompass multiple locomotor subfunctions, with a particular emphasis on the EPA Walker platform.This will enable stable and robust locomotion in the sagittal plane with prospects for expanding to three-dimensional movement. We believe that the successful harnessing of the potential of EPA design will pave the way for the development of innovative and versatile robotic systems that can mimic the agility, robustness, and efficiency of human locomotion.

## Figures and Tables

**Figure 1 biomimetics-09-00164-f001:**
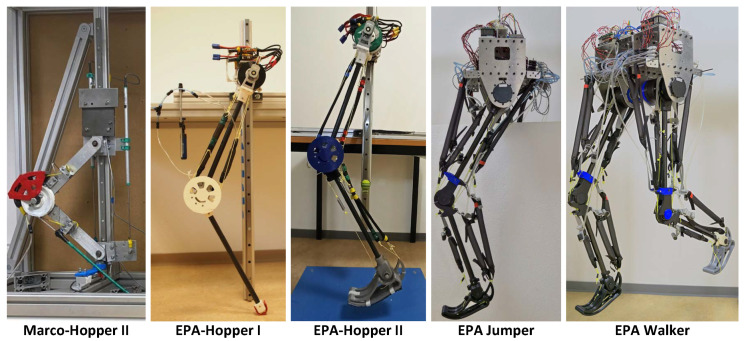
Evolutionary progression of the EPA Robot Series: MARCO-Hopper II began with a 1D hopping motion and later evolved into EPA-Hopper I, a human-sized, two-segmented leg with hip and knee joint motors and antagonistic PAMs. From this, EPA-Hopper II was designed to additionally include a 3D-printed foot, ankle-extensor PAM, and ankle-flexor spring. The iteration continued with the EPA Jumper, which incorporated a trunk and additional PAMs. This progression culminated in EPA Walker, the final bipedal iteration featuring four motors and a total of 18 PAMs for advanced mobility.

**Figure 2 biomimetics-09-00164-f002:**
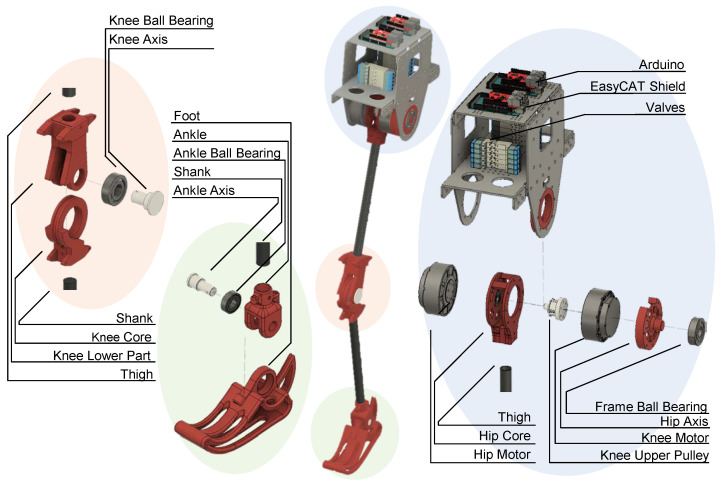
Explosive CAD visualization for the latest iteration of the EPA leg.

**Figure 3 biomimetics-09-00164-f003:**
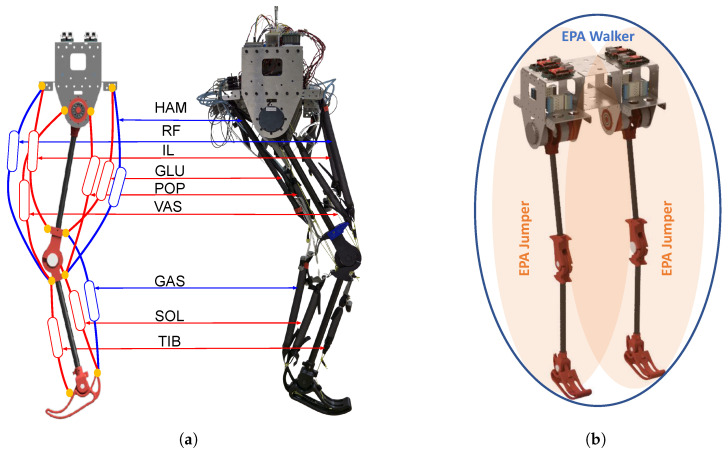
(**a**) Integration of PAMs into the robot, with the attachment points and configuration of PAMs on the robotic system being showcased. The robotic leg is equipped with six mono-articular PAMs (depicted in red) and three bi-articular PAMs (depicted in blue). The represented muscles are iliopsoas (IL), rectus femoris (RF), gluteus maximus (GL), hamstring (HAM), vastus (VAS), popliteus (POP), tibialis anterior (TIB), soleus (SOL), and gastrocnemius (GAS). (**b**) Extension from EPA Jumper to EPA Walker.

**Figure 4 biomimetics-09-00164-f004:**
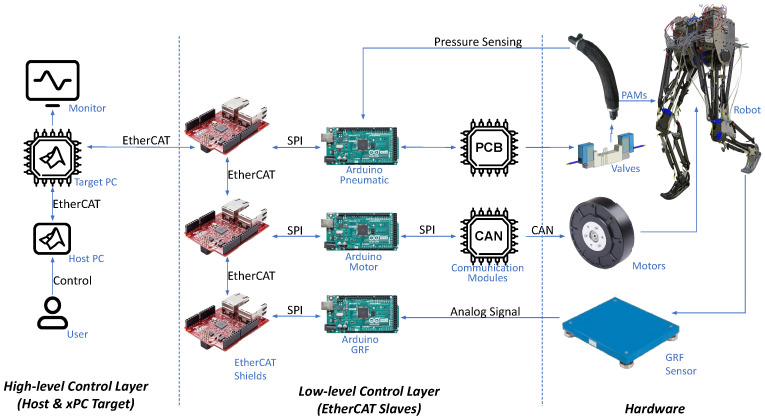
Communication network and its layers: *high-level control layer*, where control and monitoring occur within the Simulink Real-Time environment; *low-level control layer* with boards facilitating communication with the robot’s actuators and sensors; and *hardware*, encompassing the physical components of the robot. Icons and photos can be found in [66].

**Figure 5 biomimetics-09-00164-f005:**
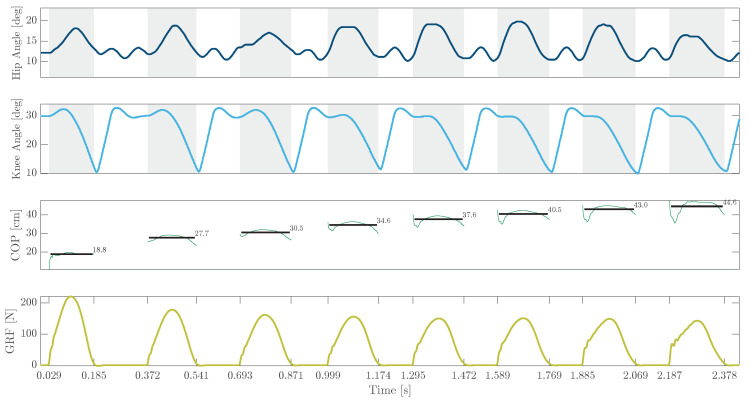
Preliminary hopping results of EPA Jumper using the FMC controller. The first two rows depict hip and knee angles throughout the hopping experiment, with gray-shaded backgrounds indicating the stance phase. The third row presents the center of pressure on the force plate with average positions highlighted at each step. The final row illustrates the ground reaction force measured by the force plate. A video of the experiment is available on [68].

**Figure 6 biomimetics-09-00164-f006:**
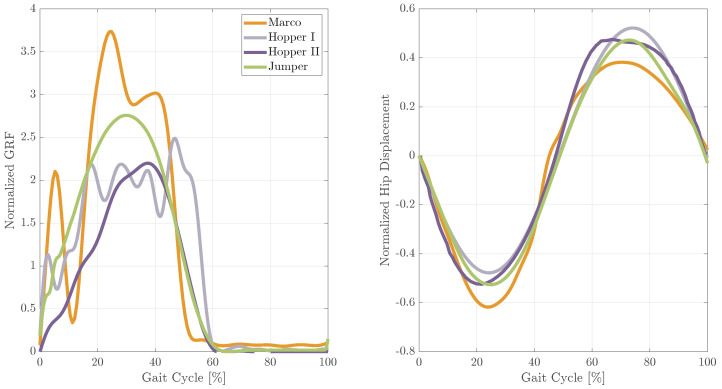
Representative hopping experiments conducted with MARCO-Hopper II, EPA-Hopper I, EPA-Hopper II, and EPA Jumper. Each graph depicts a single sample hop, starting at touchdown (0%). (**a**) Normalized GRF calculated by dividing the GRF by the corresponding robot weight and the (**b**) normalized hip displacement. A video of the robots’ hopping is available on [68].

**Table 1 biomimetics-09-00164-t001:** Comparison of pneumatic artificial muscles (PAMs) and electric motors (EMs) as actuators for robots.

Feature	PAM	EM
Output	Force	Torque
Mathematical Model	Non-Linear	Fairly Linear
Control Precision	Low	Very High
Intrinsic Compliance	Very High	Very Low
Power Density	High	Medium
Efficiency	High	High
Range of Motion	Medium	High
Dynamic Response	Slow	High
Similarity to Human Actuators	High	Very Low
Temperature Working Range	Low	High
Noise	High	Very Low
Cost	Very Low	Low
Size and Weight	Very Low	Low
Adaptability	High	Low

**Table 2 biomimetics-09-00164-t002:** List of mechanical and electrical components in the one-legged robot and trunk structure.

Component	Model	Amount
Electric Motors	GIM8115-9	2
Valves	SMC SYJ3320	9
Pressure Sensors	SMC PSE530	9
Force Plate	Kistler Type 9260AA	1
Ankle Ball Bearing	6202-2Z	1
Knee and Trunk Ball Bearing	6304-RS	2
Knee Motor Ball Bearing	6001-RS	1
Micro-controllers	Arduino Due	1
-	Arduino Mega 2560 Rev 3	2
CAN Bus	MCP2515	1
Valve Activation PCB	Custom Made	1
Ethernet Modules	EasyCat Shield	3
Target PC		1
Host PC	-	1
PLA	-	-
Metal Trunk	-	-
Dyneema Rope	-	-
Rope Tensioner	-	1
PAMs	Self-made	9
Emergency Switch	-	1
24 V LiPo Battery	GEA500012S60E	1
12 V Power Supply	-	1

**Table 3 biomimetics-09-00164-t003:** Comparison between the length, weight, and inertia of EPA Walker with humans. Segments’ length and weight were normalized to the full height and whole weight of the human and robot, respectively. The ratio of inertia (between the EPA Walker and the human model) was computed for comparative analysis.

	Segment Length [-]		Weight [-]		Inertia (Proximal) [kg·m^2^]		
	Human	EPA Walker	Human	EPA Walker	Human	EPA Walker	Ratio [-]
Foot	0.152	0.098	0.015	0.016	0.026	0.002	15.6
Shank	0.246	0.247	0.047	0.039	0.128	0.038	3.4
Thigh	0.245	0.286	0.100	0.100	0.286	0.075	3.8
Trunk	0.288	0.141	0.678	0.690	5.850	0.168	34.8

**Table 4 biomimetics-09-00164-t004:** Control parameters and PAM pressures opted for the hopping experiment with EPA Jumper.

PAM Pressure [MPa]									Flight Angles [°]		FMCK	
IL	RF	GLU	HAM	VAS	POP	TIB	SOL	GAS	Hip	Knee	$\phi_{0}$ [°]	K
0.25	0.25	0.3	0.4	0	0	0.35	0.3	0.3	10	30	10	−0.255

## Data Availability

The detailed mechanical and electrical designs of the EPA robots introduced in this study, alongside the necessary codes and software for controlling and operating these robotic systems, will be accessible within the Darmstadt TU datalib repository.

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
