# Peer review of "Design of Low-Cost Modular Bio-Inspired Electric–Pneumatic Actuator (EPA)-Driven Legged Robots"

_biomimetics, 2024, doi:10.3390/biomimetics9030164_

Round 1

Reviewer 1 Report

Comments and Suggestions for Authors

This paper presented the mechanical and electrical design of low-cost modular Electric-Pneumatic Actuator (EPA) for legged robots. Overall, the paper is well written and has solid contribution. However, several minor issues should still be addressed to further improve the quality of the manuscript. Below are some comments for the authors to consider:

1. It was mentioned several times in the paper that many parts of the EPA were fabricated using low-cost 3D printing technology. In my opinion, the authors should also provide the model name of the used 3D printer and the printing material in order to help the readers better understand the low-cost manufacturing concept. 

2. The developed EPA concept is very interesting. The authors are recommended to provide some supplementary videos to demonstrate the walking performance the EPA-driven legged robots, such as EPA Jumper and EPA Walker.

3. The literature study part could be further extended. Currently, many studies also developed underactuated compliant leg structures to achieve adaptable locomotion performance of legged robots (see the references below). From this perspective, the authors are also recommended to mention those work in the Introduction section as important state of the art. Below are some related work on underactuated compliant leg structures:

“Design of topology optimized compliant legs for bio-inspired quadruped robots”. https://doi.org/10.1038/s41598-023-32106-5

"Crab-inspired compliant leg design method for adaptive locomotion of a multi-legged robot". https://doi.org/10.1088/1748-3190/ac45e6

Reviewer 2 Report

Comments and Suggestions for Authors

Based on the previous successful work, this work designs the EPA jumper and EPA walker platforms that are closer to the human leg in weight and inertia distributions by increasing the structural complexity and more adequate EPA actuators, enabling continuous jumping actions at a low-cost and without increasing the complexity of the control strategy.

It shows that EPA has the characteristics of high power density, precise control and adjustable compliance in the bionic actuator. It can provide an experimental platform for studying the interaction between body morphology and control strategies.

This manuscript describes the iterative and optimization process for the development of a legged robot using the EPA, and details the design of the latest iteration, 

the EPA jumper, including the overall scheme of mechanical design, circuit design, and control design, which has implications for similar designs.

In general, the work content in this manuscript is rich, and the hardware and software design of the legged robot is complete.  It is suggested to increase the detail of the work in the following aspects.

1. There is too little detail on how to extend an EPAjumper into a walker in terms of structural design and control, and it is recommended to add a description of this work, the walker and the jumper should not be introduced side by side if it is not the focus of this article.

2. The hopping test in this manuscript is preliminary, the robot's motion is confined to the sagittal plane, and the walker's experiment is not carried out. The study on the motion subfunctions such as standing, swinging, and balance is not experimentally discussed. It is expected that further experiments will verify the flexibility, adaptability, and robustness of this scheme and make more discussions.

Comments on the Quality of English Language

Minor editing of English language required

Reviewer 3 Report

Comments and Suggestions for Authors

In this paper, the authors designed low-cost modular bioinspired electric-pneumatic

actuator (EPA) driven legged robots. The actuation and control of the developed legged robots were investigated. This work is of significance for assistance of human locomotion. However, there are still several concerns, as listed below, to be tackled.

1.     As claimed by the authors, the integration of EMs and PAMs address the shortcomings of each actuation system. Here, I suggest the authors to discuss how the shortcomings are avoided.

2.     In Line 173, ϕ0 represents some angle, rather than lenth.

3.     In Section 2.4.5, what is the difference between quasi-drive and drive?

4.     Figure 2 should be a table, not a figure.

Reviewer 4 Report

Comments and Suggestions for Authors

They did a very nice job. Here are my comments 

1) What is the significance of utilizing physical test benches in exploring locomotion mechanisms beyond simulation and modeling?

2) How do cost-effective modular robotic platforms contribute to validating hypotheses about real-world applications of locomotion?

3) What are the main features of the Electric-Pneumatic Actuation (EPA) robot series introduced in this study?

4) Can you explain the mechanical and electrical designs of the EPA Jumper and EPA Walker robots?

5) How do these robots replicate human weight and inertia distributions, and what is the significance of co-actuation through electrical motors and pneumatic artificial muscles?

6) What opportunities do these low-cost modular platforms offer for studying locomotor subfunctions and investigating different actuation schemes in tasks such as hopping and walking?

Round 2

Reviewer 4 Report

Comments and Suggestions for Authors

Nice job!